# An In Vitro Comparison of Elastoplastic and Viscoelastic Behavior of Dental Composites with Reversible Addition–Fragmentation Chain Transfer-Mediated Polymerization

Nicoleta Ilie

Department of Conservative Dentistry and Periodontology, University Hospital, LMU Munich, Goethestr. 70, D-80336 Munich, Germany; nilie@dent.med.uni-muenchen.de; Tel.: +49-89-44005-9412

**Abstract:** Reversible addition–fragmentation chain transfer (RAFT)-mediated polymerization has been implemented in commercially available bulk-fill dental composites, with the idea of either optimizing polymerization at depth, while providing sufficient opacity, or reducing exposure time. The elastoplastic and viscoelastic behavior of the materials pursuing both ideas are described comparatively in connection with the microstructure of the materials and artificial aging. A 3-point bending test was followed by reliability and fractographical analyses. The elastoplastic and viscoelastic behavior was monitored with an instrumented indentation test equipped with a DMA-module at various frequencies (0.5–5 Hz). Data reveal that the similarity in filler loading is reflected in similar elastic moduli. Increased strength was offset by higher plasticity and creep and was related to microstructure. Aging showed a significantly stronger influence on material behavior than differences in composition. The elastoplastic parameters of both materials deteriorate as a result of aging, but to a material-specific extent. Aging has a strong influence on elastic material behavior, but very little on viscous material behavior. The parameter that is most sensitive to aging is damping behavior. Detailed laboratory characterization indicates comparable in vitro behavior with clinically successful materials.

**Keywords:** RAFT polymerization; resin-based composites; strength; Weibull; dynamic-mechanical analysis; viscoelasticity

## 1. Introduction

Modern resin-based composites (RBC) have now improved many of their early deficiencies. Among others, polymerization shrinkage stress was held responsible when poor clinical behavior such as marginal discoloration, gaps, or even secondary caries occurred [1], while mechanical stability and biocompatibility are considered prerequisites for the long-term clinical success of a restoration [2]. Remedies such as increasing the molecular size of the monomers or decreasing crosslink density [3], the development of systems that can expand during polymerization, such as silorane and oxirane [4], or the optimization of the filler systems have been enriched in modern times with the introduction of new polymerization mechanisms such as the reversible addition–fragmentation chain transfer (RAFT) radical polymerization [5–7].

Although the RAFT methodology was developed more than 20 years ago to reduce the disadvantages of free radical polymerization—identified in the broad molecular weight distribution and limited control over the final polymer architecture and the end group functionality [8]—the process was applied very late in dental RBCs; the first patent was only published in 2015 [6]. This was probably due to the difficulty in finding a suitable RAFT agent [9], which is basically the only necessarily addition to the chemical composition of conventional cured methacrylate-based RBCs via radical polymerization [8,10] since such compounds add odor and/or color, both of which are incompatible with a

dental restorative material. It is likely that with the advent of bulk-fill materials that are polymerized in 4–5 mm increments in one step, there has been increased pressure to ensure adequate in-depth polymerization while also emphasizing aesthetics. The latter has been an issue to some extent, as many of the earlier (but also some current) bulk-fill RBCs use the higher translucency to enable more light in depth, which involves either increasing the size of the filler to reduce the filler–matrix interface where light is scattered, or reducing the filler amount [11]. However, highly translucent materials in large cavities may not adequately mask tooth discoloration or may appear greyish in a clinical situation. Therefore, for aesthetic reasons, when using large RBC increments to speed up the restoration process, alternative mechanisms must be considered to ensure adequate polymerization throughout the restoration. One of the proposed mechanisms for this is the RAFT-mediated polymerization [6].

The RAFT polymerization process operates on the principle of reversible chain transfer, with two additional steps—a pre-equilibrium and a main-equilibrium—superimposed on a conventional free-radical polymerization scheme [8]. Both polymerization mechanisms start with the initiation of polymerization followed by the propagation step that involves increasing chain length after radical transfer from the initiator molecules to the monomer units. In a RAFT polymerization, a subsequent step occurs—the pre-equilibrium—in which the propagating radical reacts with the RAFT agent to form an intermediate radical, which can then undergo a fragmentation reaction, either yielding back the reactants or releasing an initiating leaving group radical under the concomitant formation of a polymeric compound [8]. A re-initiation of the polymerization follows, as the leaving group radical can react with another monomer species, allowing another active polymer chain to be started. Recurring RAFT events then establish equilibrium between dormant and living chains, characterizing the main-equilibrium step. Similar to the conventional radical polymerization, the RAFT-mediated polymerization is terminated by the reaction of active chains via bi-radical termination and the formation of chains that cannot react further (dead polymers) [8].

The aim of this study was to offer a comparative evaluation of the elastoplastic and viscoelastic behavior of commercially available dental RBCs with reversible addition–fragmentation chain transfer-mediated polymerization.

The null hypotheses tested were that RAFT-mediated RBCs behave similarly with respect to (a) strength, elastic modulus, beam deflection at fracture, reliability, and fracture pattern; (b) elastoplastic behavior; (c) viscoelastic behavior; and (d) aging under clinically simulated conditions.

## 2. Materials and Methods

The flexural strength, FS, flexural modulus, E, and beam deflection, $\varepsilon$, were determined in a three-point bending test at 24 h post-polymerization. The fractured specimens were then all analyzed fractographically. The quasi-static and viscoelastic behavior of the analyzed RBCs was monitored by an instrumented indentation test equipped with a DMA module (FISCHERSCOPE® HM2000, Helmut Fischer, Sindelfingen, Germany) at 24 h post-polymerization and 3-months artificial aging. Material details are summarized in Table 1. A violet-blue LED (Light-Emitting Diode) LCU (Light Curing Unit) (Bluephase® Style, Ivoclar Vivadent, Schaan, Liechtenstein; $1391.3 \pm 5.8$ mW/cm$^2$) was used for polymerization.

### 2.1. Three-Point Bending Test

A three-point bending test was performed according to NIST No. 4877 with a distance of 12 mm between the supports [12] and ISO 4049:2009 [13]. For this purpose, 40 (n = 20) specimens were produced by compressing the material between two glass plates with polyacetate sheets in between, which were separated by a white polyoxymethilen mould with an inner dimension of 2 mm × 2 mm × 18 mm. The specimens were light cured as specified by the manufacturer, which was 10 s for TPF and 20 s for FO, and stored in distilled water at 37 °C immediately after demolding for 24 h. Specimens were loaded until

fracture in a universal testing machine (Z 2.5, Zwick/Roell, Ulm, Germany) at a crosshead speed of 0.5 mm/min. The universal testing machine measured the force during bending as a function of the deflection of the beam. The flexural modulus was calculated from the slope of the linear part of the force-deflection diagram. The deflection at fracture $\varepsilon$ was also recorded.

**Table 1.** RAFT RBCs: Abbreviation (code), brand, manufacturer, shade, LOT, and composition, as indicated by the manufacturer.

| Code | Material Manufacturer | Shade | Exposure | LOT | Monomer | Filler | |
|---|---|---|---|---|---|---|---|
| | | | | | | Composition | wt/Vol% |
| FO | Filtek™ One 3M, St. Paul, MN, USA | A2 | 20 s | N963171 | UDMA, DDDMA, AUDMA, AFM | $SiO_2/ZrO_2$ $YbF_3$ | 76.5/58.5 |
| TPF | Tetric PowerFill Ivoclar Vivadent, Schaan, Liechtenstein | $^{IV}$A | 10 s | Y31685 | Bis-GMA, Bis-EMA, UDMA, DCP, PO-Bis-GMA [5] | $BaO\text{-}Al_2O_3\text{-}$ $SiO_2$, $SiO_2/ZrO_2$, $YbF_3$ | 77/53–54 |

Abbreviations: Bis-GMA = bisphenol A glycol dimethacrylate; Bis-EMA = ethoxylated bisphenol A dimethacrylate; UDMA = urethane dimethacrylate; DCP = tricyclodocane dimethanol dimethacrylate; DDDMA = 1, 12-dodecanediol dimethacrylate; PO-Bis-GMA propoxylated bisphenol A dimethacrylate; AFM = "addition fragmentation monomers" of unspecified composition, AUDMA = aromatic urethane dimethacrylate; $SiO_2$ = silicon oxide (silica); $ZrO_2$ = zirconium oxide; $YbF_3$ = ytterbium trifluoride; $BaO\text{-}Al_2O_3\text{-}SiO_2$ = barium aluminosilicate glass; "-" not specified.

### 2.2. Fractography Analysis

The fractography was performed with a stereomicroscope (Stemi 508, Carl Zeiss AG, Oberkochen, Germany) in order to determine the fracture pattern and fracture origin. All fractured surfaces were therefore photographed using a microscope extension camera (Axiocam 305 color, Carl Zeiss AG, Oberkochen, Germany). The origin of fracture was identified either as a volume (sub-surface) or a surface (edge, corner) defect.

### 2.3. Scanning Electron Microscopy (SEM) Evaluation

The structural appearance of the filler systems was analyzed using scanning electron microscope (SEM, Zeiss Supra 55 V P, Carl Zeiss AG, Oberkochen, Germany) operating in the electron backscatter diffraction mode. Samples of each material were prepared similarly as above (n = 3) and wet processed after 24 h of storage by means of an automatic grinder (EXAKT 400CS Micro Grinding System, EXAKT Technologies Inc., Oklahoma City, OK, USA) with gradually finer silicon carbide abrasive papers (1200, 1500, 2000, and 2400 grit). Surface preparation was completed by polishing the surface with a 1 μm diamond spray (DP-Spray, STRUERS GmbH, Puch, Austria).

### 2.4. Instrumented Indentation Test (IIT)

2.4.1. Quasi-Static Indentation Test

Randomly selected fragments (n = 5) from the 3-point bending test were either immediately exposed to the IIT or additionally stored in artificial saliva (pH 6.9; composition: 1.2 g potassium chloride, 0.84 g sodium chloride, 0.26 g di-potassium hydrogen phosphate, and 0.14 g calcium chloride dihydrate per 1000 g of water) in a dark environment at 37 °C for 3 months. The artificial saliva was renewed weekly. Previous to each measurement, specimens were wet-ground with silicon carbide abrasive paper as described above and then polished with a diamond suspension (mean grain size: 1 μm) for 2–3 min until the surface was shiny (EXAKT 400CS Micro Grinding). Specimens were exposed to a quasi-static indentation test according to ISO 14577 [14] while employing an automated nano-indenter (FISCHERSCOPE® HM2000) equipped with an Vickers diamond tip. Three measurements were randomly performed on each sample (n = 5) and each material. Each indentation was performed force controlled by increasing the test load over 20 s from 0.4 mN to 1000 mN,

followed by maintaining the maximal force for an additional 5 s and then decreasing the force over 20 s at a constant speed. Within each load–unload cycle, the load (F) and indentation depth (h) of the indenter were continuously measured, allowing the calculation of a range of parameters that characterize the elastic and plastic deformation. The integral of the force with depth (= $\int$ Fdh) defines the total mechanical work of indentation $W_{total}$. During the indentation procedure, a part of the total mechanical work is consumed as plastic deformation work $W_{plast}$, while the rest is set free as work of the elastic reverse deformation $W_{elastic}$. The ratio of the elastic reverse deformation work of indentation ($W_{elast}$) to the total mechanical work of indentation ($W_{total}$) was then calculated, and it represents a prerequisite variable for the further DMA test ($W_{elast}/W_{total} = \mu_{IT}$). Further parameters were then determined from the load–indentation depth variation; these include the indentation modulus, $E_{IT}$, which was calculated from the slope of the tangent of the indentation depth curve at the maximum force. Hardness, with its plastic and elastic components, was calculated by evaluating the impression created during the indentation. For this purpose, the projected indenter contact area ($A_c$) was determined from the force–indentation depth curve, while considering the indenter correction based on the Oliver and Pharr model (and described in ISO 14577) [14] and a previous calibration with sapphire and quartz glass. The resistance to plastic deformation only is described by the indentation hardness ($H_{IT} = F_{max}/A_c$) and its more familiar correspondent, the Vickers hardness (HV = 0.0945 × $H_{IT}$). The universal hardness (or Martens hardness = $F/A_s(h)$) was calculated by dividing the test load by the surface area of the indentation under the applied test load ($A_s$), and it characterizes both plastic and elastic deformation. Creep was calculated from changes in indentation depth during the 5 s of maintaining maximal indentation force during the indentation process described above. The indentation depth at maximal force is also indicated ($h_{max}$).

### 2.4.2. Dynamic Mechanical Analysis (DMA)

The DMA test used a low-magnitude oscillating force (10 different frequencies in the range 0.5–5 Hz) that was superimposed onto a quasi-static force of 1000 mN. The oscillation amplitude was set at 5 nm, so that the sample deformation kept within the linear viscoelastic regime. Three randomly chosen indentations have been performed per each specimen (n = 5), amounting 15 individual measurements per RBC brand and aging conditions. Within each indentation, ten measurements were performed for each of the frequencies used. For the used frequency (0.5 Hz; 0.7 Hz; 0.9 Hz; 1.1 Hz; 1.4 Hz; 1.8 Hz; 2.3 Hz; 3.0 Hz; 3.9 Hz; and 5.0 Hz), the force oscillation generates oscillations on the displacement signal with a phase angle δ. The sinusoidal response signal was then separated into a real part and an imaginary part, representing the storage (E′) and the loss moduli (E″), respectively. E′ is a measure of the elastic response of a material behavior, whereas E″ characterizes the viscous material behavior. The quotient E″/E′ is defined as the loss factor (tan δ) and is a measure of the material damping behavior.

The indentation hardness $H_{IT}$ was determined along with the above described viscoelastic parameters as a measure of the resistance to plastic deformation and was calculated as the ratio between the applied load and the contact area ($H_{IT} = F_{max}/A_p$) [14] at each frequency.

### 2.5. Statistical Analyses

The distribution of the variables was tested with the Shapiro–Wilk procedure. All variables were normally distributed enabling using a parametric approach. Multifactor analysis of variance was applied to compare the parameters of interest (flexural strength, FS; flexural modulus, E; beam deflection at fracture $\varepsilon$; fracture mode; Martens hardness HM; Vickers hardness, HV; indentation hardness, $H_{IT}$; indentation modulus, $E_{IT}$; elastic indentation work, $W_e$; total indentation work, $W_t$; ratio of the elastic to the total indentation work, $\mu_{IT} = W_e/W_t$; creep; maximal indentation depth, $h_{max}$; storage modulus, $E'$; loss modulus, $E''$; loss factor, tan $\delta$) among analyzed materials, aging conditions and frequencies. The results were compared using a Student's *t*-test, multiple-way analysis of variance

(ANOVA), and Tukey honestly significant difference (HSD) post hoc test ($\alpha$ = 0.05), using an alpha risk set at 5%. A multivariate analysis (general linear model) assessed the effect strength of parameters RBC, aging, and frequency, as well as their interaction terms on the analyzed properties. The partial eta-squared statistic reported the practical significance of each term, based on the ratio of the variation attributed to the effect. Larger values of partial eta-squared ($\eta_P{}^2$) indicate a greater amount of variation accounted for by the model (SPSS Inc. Version 29.0, Chicago, IL, USA).

Flexural strength data were additionally characterized by Weibull analysis to determine material reliability. A common empirical expression for the cumulative probability of failure *P* at applied stress $\sigma$ is the Weibull model [15] is as follows:

$$P_f(\sigma_c) = 1 - \exp\left[-\left(\frac{\sigma_c}{\sigma_0}\right)^m\right],$$

where $\sigma_c$ is the measured strength, *m* the Weibull modulus, and $\sigma_0$ the characteristic strength, defined as the uniform stress at which the probability of failure is 0.63. The double logarithm of this expression gives $\ln\ln\frac{1}{1-P} = m\ln\sigma_c - m\ln\sigma_0$. By plotting $\ln\ln(1/(1-P))$ versus $\ln\sigma_c$, a straight line results with the upward gradient, *m*; whereas the intersection with the x-axes gives the logarithm of the characteristic strength [15].

## 3. Results

### 3.1. Three-Point Bending Test and Fractography Analysis

The parameters measured in the 3-point bending test are summarized in Table 2 and Figure 1. A Student's *t*-test evidenced statistically similar E values for both materials (*p* = 0.072), but significantly higher FS (*p* < 0.01) and beam deflection (*p* < 0.01) in FO compared to TPF. The Weibull analysis identified slightly higher reliability in FO.

**Table 2.** Three-point bending test; flexural strength, FS, with Weibull parameters (m with standard error in parenthesis; characteristic strength, $\sigma_0$, which is the strength at a probability of failure P of 63.2%; and R-squared ($R^2$) values); flexural modulus, E; and beam deflection at fracture, $\varepsilon$ (mean and standard deviation SD). Values denoted by the same superscript are statistically similar.

| RBC | FS, MPa | | Weibull Parameters | | | E, GPa | | $\varepsilon$, % | |
|---|---|---|---|---|---|---|---|---|---|
| | Mean | SD | m | $\sigma_0$ | $R^2$ | Mean | SD | Mean | SD |
| FO | 181.9 [a] | 12.4 | 16.9 [a] (0.34) | 187.7 | 0.94 | 7.9 [a] | 0.8 | 2.7 [a] | 0.3 |
| TPF | 124.9 [b] | 9.7 | 15.6 [b] (0.21) | 129.1 | 0.97 | 7.4 [a] | 0.7 | 1.8 [b] | 0.2 |

Fractography Analysis

The fracture mode analysis (Figure 2) evidenced failures initiated from surface defects (edge, 47.5%; and corner, 7.5%) as the most frequent type of failure (55%), with volume defects (sub-surface mode) amounting to 45%. The amount of failure initiated by volume defects was higher in FO compared to TPF.

### 3.2. Scanning Electron Microscopy (SEM) Evaluation

The filler system is illustrated in Figure 3, evidencing larger (up to 4 µm) and predominantly round fillers in FO compared to smaller (up to 1.5–2 µm) and predominantly edgy fillers in TPF. As the structural appearance of the filler systems was visualized by scanning electron microscopy in electron backscatter diffraction mode, pictures allowed a distinction to become apparent between the different chemical compositions of the fillers.

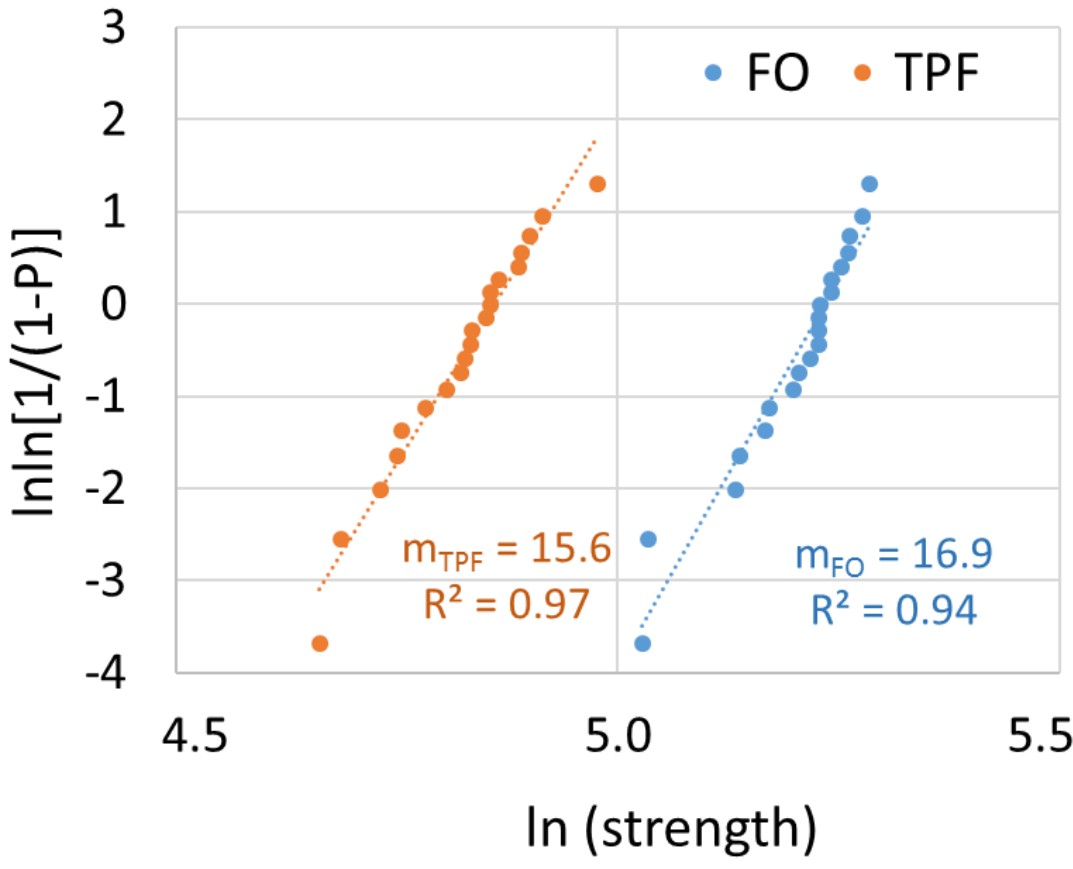

**Figure 1.** Weibull plot representing the empirical cumulative distribution function of strength data. Linear regression was used to numerically assess goodness of fit and estimate the parameters of the Weibull distribution.

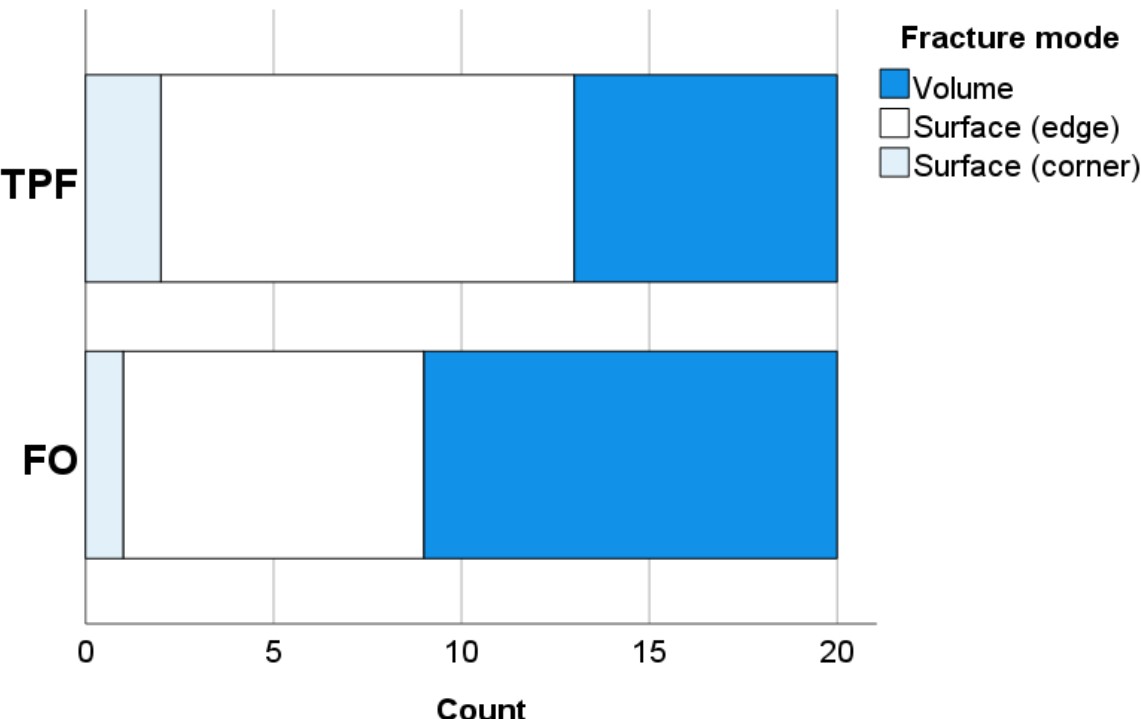

**Figure 2.** Fracture mode distribution among analyzed RBCs.

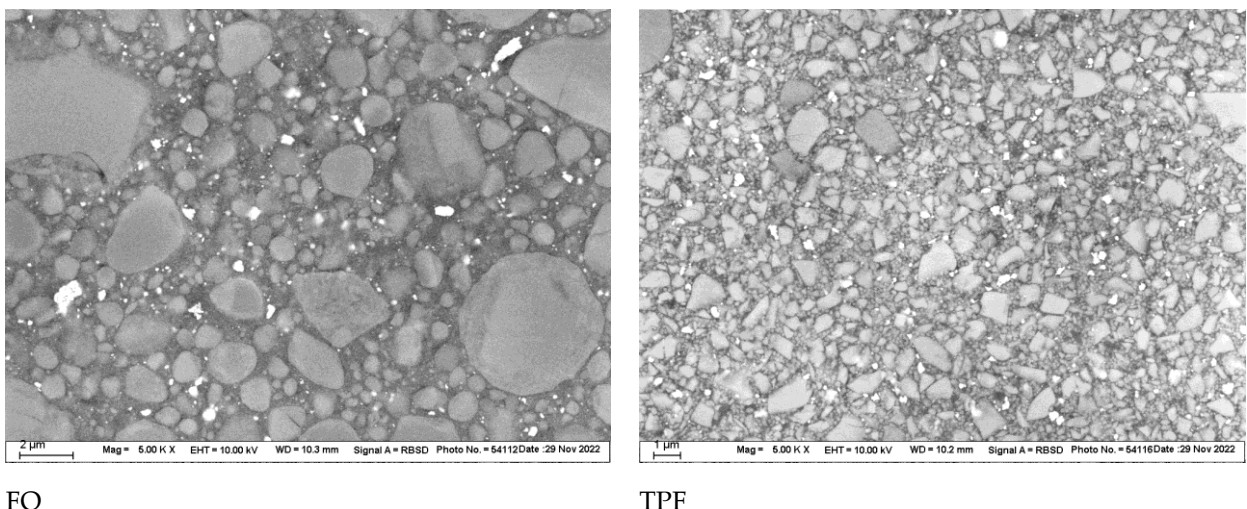

FO                                                                TPF

**Figure 3.** SEM images of the filler system in FO and TPF.

### 3.3. Instrumented Indentation Test (IIT): Quasi-Static Approach

A multifactorial analysis shows a significant ($p < 0.001$) influence of the analyzed parameters RBC and aging, with the exception of the parameter indentation modulus, which was not influenced by the RBC ($p = 0.120$). Aging evidenced a significantly stronger influence on the measured parameters than RBC (higher $\eta_P^2$ values, Table 3), except for the parameter creep, where the opposite was observed. The binary interaction product RBC × aging was also significant and strong, with $\mu_{IT}$ being the only parameter that remained unaffected ($p = 0.297$).

**Table 3.** Quasi-static parameters. (**a**). Effect strength of the factors aging and RBC and their interaction product on the measure parameters of the quasi-static IIT. Partial eta-squared values $\eta_P^2$ are indicated when the effect was significant ($p < 0.001$); n.s. is reported if the effect was not significant; HM = Martens hardness, HV = Vickers hardness; $E_{IT}$ = indentation modulus; $\mu_{IT} = W_e/W_t$; $W_e$ = elastic indentation work; $W_t$ = total indentation work; Creep; $h_{max}$ = maximal indentation depth. (**b**). Mean values and standard deviation (SD) of the quasi-static IIT parameters at measuring conditions.

**(a)**

| Parameter | HM | HV | $E_{IT}$ | $\mu_{IT}$ | $W_e$ | $W_t$ | Creep | $h_{max}$ |
|---|---|---|---|---|---|---|---|---|
| Aging | 0.925 | 0.927 | 0.914 | 0.515 | 0.670 | 0.924 | 0.556 | 0.928 |
| RBC | 0.398 | 0.374 | n.s. | 0.087 | 0.244 | 0.157 | 0.821 | 0.359 |
| Aging × RBC | 0.583 | 0.532 | 0.652 | n.s. | 0.475 | 0.71 | 0.145 | 0.625 |

**(b)**

| RBC | Aging | HM, N/mm$^2$ | | HV, N/mm$^2$ | | $E_{IT}$/(1-vs$^2$), GPa | | $\mu_{IT}$, % | |
|---|---|---|---|---|---|---|---|---|---|
| | | Mean | SD | Mean | SD | Mean | SD | Mean | SD |
| FO | 24 h | 596.0 | 11.2 | 87.6 | 1.7 | 13.3 | 0.3 | 45.2 | 0.4 |
| TPF | | 604.4 | 12.9 | 88.6 | 2.0 | 13.9 | 0.3 | 45.8 | 0.8 |
| FO | 3 months | 542.4 | 11.6 | 78.5 | 1.8 | 12.3 | 0.3 | 44.0 | 0.3 |
| TPF | | 496.7 | 11.6 | 71.8 | 2.0 | 11.5 | 0.2 | 44.2 | 1.0 |

**Table 3.** *Cont.*

| RBC | Aging | $W_e$, μJ | | $W_t$, μJ | | Creep, % | | $h_{max}$, μm | |
|-----|-------|-----------|---|-----------|---|----------|---|---------------|---|
| | | **Mean** | **SD** | **Mean** | **SD** | **Mean** | **SD** | **Mean** | **SD** |
| FO | 24 h | 1.2 | 0.01 | 2.7 | 0.04 | 4.7 | 0.07 | 8.2 | 0.08 |
| TPF | | 1.2 | 0.02 | 2.6 | 0.03 | 4.2 | 0.14 | 8.1 | 0.09 |
| FO | 3 months | 1.2 | 0.02 | 2.8 | 0.03 | 4.9 | 0.09 | 8.6 | 0.09 |
| TPF | | 1.3 | 0.03 | 2.9 | 0.02 | 4.5 | 0.08 | 9.0 | 0.10 |

### 3.4. Instrumented Indentation Test (IIT): Dynamic Mechanical Analysis (DMA)

The pattern of variation of the measured property with frequency was similar for both materials and both aging conditions within a parameter. Storage modulus, E′, loss modulus, E″, and loss factor decrease exponentially with increasing frequency, while only a slight increase with frequency can be observed for the indentation hardness $H_{IT}$ (Figure 4). $H_{IT}$ and E′ curves plateau earlier (1.1 Hz) than loss modulus and loss factor (3 Hz). The loss factor differentiates the effects of aging more discriminatively than the loss modulus, while the differences decrease with frequency.

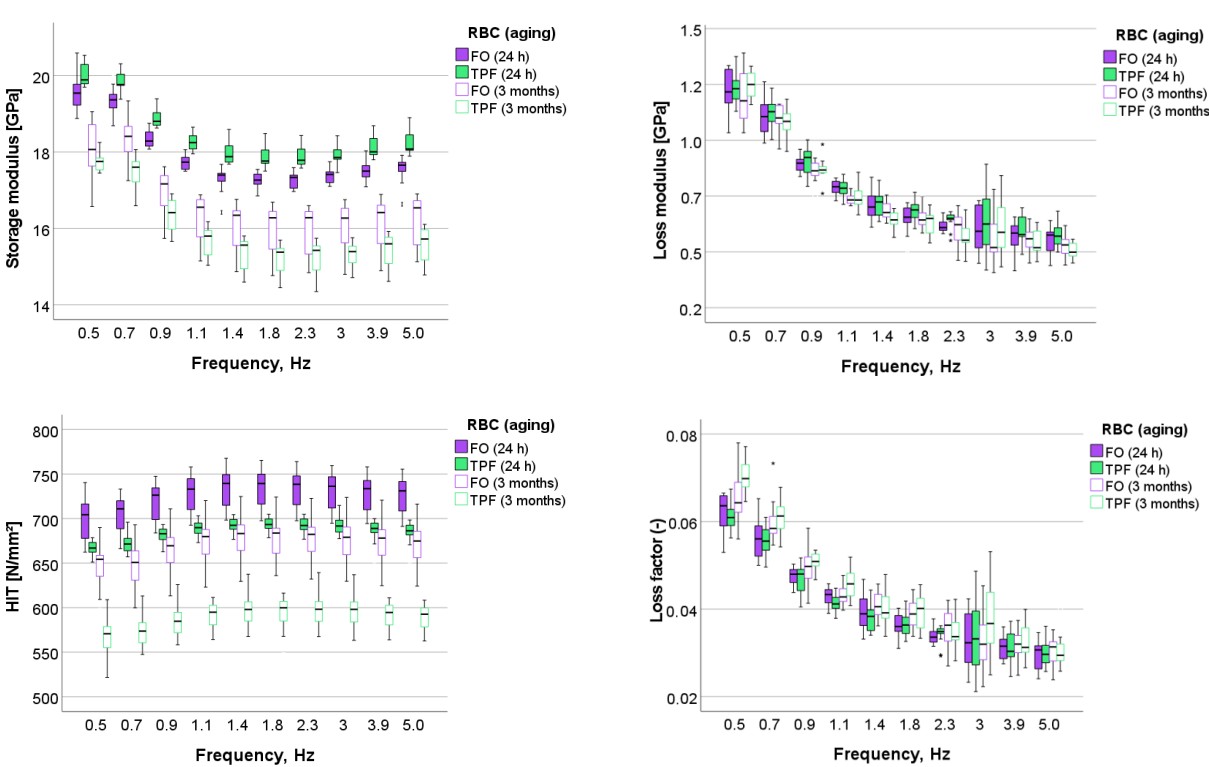

**Figure 4.** Dynamic mechanical analysis: variation of the storage and loss modulus, indentation hardness, and loss factor over the frequency range 0.5–5 Hz for the analyzed materials and aging conditions.

Compared to FO, TPF has a slightly higher storage modulus when stored for 24 h, but a significantly lower indentation hardness. The values for both materials decrease after 3 months of storage (more so for TPF than for FO). Differences in loss modulus are very small, both between materials and through aging. In contrast, the loss factor distinguishes between the material performance after aging; the better, the lower the frequency.

A multifactorial analysis reveals the significant ($p < 0.001$, Table 4) effect of aging on all parameters, which is very high on the parameters characterizing the elastic material behavior (E′ and $H_{IT}$) and still significant, but very low on the parameters characterizing the viscous material behavior (E″ and tan δ). The RBC only has a significant impact on

$H_{IT}$. In contrast, frequency significantly ($p < 0.001$) affected all parameters, while the effect was very strong (high $\eta_P^2$ values) on E′, E″, and tan δ, but small on $H_{IT}$. From the binary combination of effects, apart from E″, only aging × RBC exerted a significant influence on the measured properties. The ternary combination of the effects was not significant.

**Table 4.** Effect strength of aging, RBC, frequency, and their interaction products on the measure parameters of the DMA IIT. Partial eta-squared values $\eta_P^2$ are indicated when the effect was significant ($p < 0.001$); n.s. is reported if the effect was not significant.

| Parameter | E′ | E″ | $H_{IT}$ | tan δ |
|---|---|---|---|---|
| Aging | 0.833 | 0.06 | 0.780 | 0.081 |
| RBC | n.s. | n.s. | 0.681 | n.s. |
| Frequency | 0.784 | 0.907 | 0.203 | 0.875 |
| Aging × RBC | 0.351 | n.s. | 0.214 | 0.011 |
| Frequency × RBC | n.s. | n.s. | n.s. | n.s. |
| Aging × Frequency | n.s. | n.s. | n.s. | n.s. |

## 4. Discussion

Given the rarity of RAFT-mediated polymerization applied in dental materials to date, the present study aimed to directly compare the currently commercially available materials and to evaluate the effect of their compositional specifics on elastoplastic and viscoelastic behavior.

The main advantage of the RAFT polymerization in dental materials is that the chemical composition of the monomer, filler, and photoinitiator systems does not need to be changed. The RAFT-polymerized RBCs are thus compatible with regular RBCs and adhesive systems and can be cured with normal light curing devices that every dentist already has in their practice. In fact, the practitioner does not register any differences in the way the material needs to be applied, cured, and handled clinically.

Despite the above advantages, RAFT polymerization has so far only been incorporated into two dental RBCs, both of which act as bulk-fill materials. One of these materials (Filtek™ One, 3M, St. Paul, MN, USA) pursued the idea of optimizing polymerization in depth in a sufficiently opaque material, while the second material (Tetric PowerFill, Ivoclar Vivadent, Schaan, Liechtenstein) aimed at faster and more efficient polymerization, namely at 3 s exposure. In fact, both approaches turned out to be efficient, as it was found that changing the polymerization mechanism to RAFT-mediated polymerization allows for adequate in-depth curing in both materials [10,16]. It was further shown that RAFT-mediated polymerization induces in Tetric PowerFill at 3 s curing with high irradiance comparable in-depth mechanical properties and degree of conversions to the traditional free-radical polymerization-cured RBC with a nearly equivalent chemical composition [10]. Both approaches have been realized through the development of a specific RAFT agent, which is an addition–fragmentation monomer in Filtek™ One and a β-allyl sulfone— specifically synthesized for dental materials—in Tetric PowerFill [5]. The β-allyl sulfone belongs to a class of compounds previously shown to be potent reagents in methacrylate systems [9].

While tests such as three-point bending or hardness measurements are common practice, the viscoelastic behavior of dental materials has only rarely been assessed [17,18]. However, the presence of polymer in RBCs, in addition to the friction occurring at the interphase boundary of fillers, polymer, or inherent defects such as voids [19], leads to pronounced viscoelastic behavior [17,18,20]. This behavior involves a time-dependent recovery during loading [19] that is ignored by traditional static testing, making it difficult to relate mechanical behavior to the materials' microstructure [21]. The quasi-static methodologies were therefore extended in the present study to include DMA investigations to assess viscoelastic material behavior using oscillating (sinusoidal) components. Moreover,

the frequency range of 0.5–5 Hz was chosen to comprise human chewing activity, which was quantified at 0.94 Hz to 2.17 Hz [22].

The analyzed materials are both methacrylate-based, although the chemical composition of the organic matrix differs. FO is exclusively urethane dimethacrylate-based, while TPF consists of a blend of bis-GMA/UDMA monomers. In contrast, the filler systems are quite comparable in terms of filler loading, with similar filler weights and only slightly higher (4.5%) filler volume in FO. Although the proportion of inorganic fillers has a direct impact on the elastic modulus [23], the above-mentioned difference in filler volume was small enough not to be noticeable in either the macroscopically measured elastic modulus or the microscopically measured indentation modulus 24 h post-polymerization; the effect manifested itself later, after aging, by a greater drop in indentation modulus and hardness in TPF compared to FO, which requires a more detailed consideration of the microstructure, geometry, and shape of the fillers. In fact, the SEM analysis shows large differences in the shape and size of the fillers; in FO, they were larger and predominantly round, while in TPF, the fillers were smaller and predominantly edgy. Apparently, both materials contain a small amount of $YbF_3$, which appears as very light white, homogeneously dispersed nanoparticles. In addition, both materials contain round or almost round $SiO_2/ZrO_2$ fillers. They represent the predominant type of fillers in FO, while their proportion in TPF is small and can be illustrated by the large, round light grey filler in the upper left corner of the image presented in Figure 3. Some fillers of the same composition can be seen in the center of the same image; their shape is not perfectly round but is less angular than the rest of the fillers in TPF. In contrast to the $SiO_2/ZrO_2$ fillers, the majority of the fillers in TPF appear lighter and can be attributed to the barium silicate glass fillers, with irregular, edgy geometry. The difference in colour of the fillers is due to the fact that Ba has a much higher atomic order than Zr and the fillers containing it appear lighter as a result. By this observation, it can be inferred that the smaller fillers in TPF compared to FO imply a significantly larger filler–matrix interface, which is known to play an important role in the degradation of a composite and can be held responsible for the behavior of the two materials after aging. This behavior is in line with previous studies that indicate a slight decrease in properties such as hardness and modulus of elasticity after aging in RBCs with nanoparticles and agglomerations of nanoparticles, while being characterized by very good mechanical properties and excellent reliability [17,24]. On a positive note, the reliability of both materials is high (Weibull analysis), with a slight—at the limit of the significance—superiority of FO. The small difference between both materials may to some extent be related to differences in porosity, as pores may act as defects able to initiate fracture. Porosity is inevitable in both the uncured monomer paste and the cured polymer, as additional porosity can be added during sample preparation. In fact, it was found that the average closed porosity in FO (0.002%) compared to TPF (0.007%) was 3.5 times lower in the uncured material and increased in both materials after curing while the difference was magnified (0.003% in FO and 0.013% in TPF) [25]. As pores are considering defects that may initiate fracture, their lower amount in FO may contribute to the slightly higher material reliability compared to TPF. In addition, it must be mentioned that the modulus of elasticity was statistically similar for both materials, but the flexural strength of FO was significantly higher. This behavior must be related to the higher plastic deformation capacity observed in FO and confirmed by the higher beam deflection, resulting in the tested beam taking longer to fail under load in the 3-point bending test. This behavior corresponds to the data measured in the quasi-static indentation test, since FO has higher creep values compared to TPF, which means higher deformation under load. Interestingly, the difference in creep between the materials remains after aging in favour of TPF, although creep increases for both materials.

To cure the materials, we opt for a single curing device and the manufacturer's recommended exposure time of 10 s for TPF and 20 s for FO. The exposure times were confirmed in a recent study using an LCU with a comparable irradiance, showing no variation in the degree of conversion (DC) measured on both the top and the bottom

of 4 mm thick specimens when TPF was cured at 10 s or 20 s. This justifies the chosen 10 s exposure in the present study. In contrast, DC in FO increased significantly when exposure was increased from 10 s to 20 s [25], so the selection of the 20 s exposure in FO is also validated.

Compared to the quasi-static test, the DMA allows the complex modulus to be split into an elastic part (storage modulus, E′), which reflects the material's ability to store elastic energy associated with recoverable elastic deformation, and a viscous part (loss modulus, E″), which characterizes the dissipated energy [21]. Another material parameter that is of great importance in this context is the damping behavior of the material, which reflects its energy dissipation potential and is quantified by the loss factor (tan δ); it is, in fact, the ratio of the viscous to the elastic material response. Therefore, high tan delta values are sought in order to develop materials that are better able to dissipate mechanical stress and thus behave better under clinical situations. It is important to note that materials do not differ in terms of loss modulus, which remains constant even during aging. The increase in the damping behavior of both materials due to aging is therefore more related in both materials to the decrease in the storage modulus. A comparison of the damping behavior of FO with similarly structured materials from the same manufacturer, such as Filtek Supreme XTE, shows very similar behavior [17]. On a general note, the damping behavior of the analyzed materials is comparable to some other representative materials such as Venus and Venus Diamond (Kulzer) [24], but their values tend to be in the lower range of the tested commercially available materials [17,18]. As a general observation, and similar to other RBCs cured by a radical polymerization, both materials are better adapted to the chewing frequency of humans (0.94 Hz to 2.17 Hz [22]) than to higher frequencies. This behavior is attributed to the polymer content, its flexibility, and the time it takes for a polymer chain to adapt to the applied stress [21] because the faster a stress is applied (higher frequency), the shorter the time available for the molecules to relax and accommodate that stress. Interestingly, both materials behave very similarly in this regard, which does not allow any statement to be made about differences in flexibility between the polymer networks or different crosslinking densities.

While small differences within the analyzed materials were found in the large number of parameters measured, the differences were mostly significant, so all null hypotheses could be rejected.

## 5. Conclusions

- RAFT-mediated RBCs differ less from each other in many of the properties analyzed due to a comparable ratio of inorganic fillers to polymer matrix, but age slightly differently due to microstructural differences;
- With similar moduli of elasticity and viscous behavior, the material with higher flexibility (FO) manifests in higher flexural strength at the macroscopic scale, but also higher creep;
- Due to the small differences in their behavior, it can be expected that both materials will behave similarly in a clinical context.

**Funding:** This research received no external funding.

**Data Availability Statement:** Data is available on request.

**Conflicts of Interest:** The author declares no conflict of interest.

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
