# Peer review of "An In Vitro Comparison of Elastoplastic and Viscoelastic Behavior of Dental Composites with Reversible Addition–Fragmentation Chain Transfer-Mediated Polymerization"

_jcs, doi:10.3390/jcs7060247_

Round 1

Reviewer 1 Report

Comments to the Authors:

The objective of this study is to investigate the dental composites with reversible addition-fragmentation-chain transfer mediated polymerization: An in-vitro comparison of their elasto-plastic and viscoelastic behaviour.  Some suggestions are listed as following:

1.        The title of this manuscript is ambiguous to general readers in many aspects.  First, this manuscript claimed the term of “reversible addition-fragmentation-chain transfer mediated polymerization” only by testing the mechanical properties of 2 commercial dental composite products but without showing any mechanism of chemical reaction related with “reversible addition−fragmentation” and “chain transfer polymerization” using monomers and β-allyl 95 sulfone.  In addition, this type of writing style should only happen in author conduct your own RAFT polymerization with some modified monomers or reagents.  Or the manuscript is some kinds of review paper.

2.        Second, the term of “in-vitro comparison” usually refers to a test occurring in a laboratory involving the studying of the biological effects for microorganisms or human/animal cells in culture.  I don’t see any relevant in vitro study results in this manuscript.

3.        Third, the term of “elasto-plastic and viscoelastic behavior” should be clearly defined and classified for dental composites by testing their stress-strain curves or any other appropriate test according their definition. Elastoplasticity is a phenomenon where a material undergoes permanent deformation when the applied stress exceeds a critical value, and elastoplastic parameters are quantities that describe the elastic and plastic deformation behavior of a material.  There are many terms and data showed in the Table 3 like HM = Martens hardness, HV = Vickers hardness; EIT = indentation modulus; μIT = We / Wt; We = elastic indentation work; Wt = total indentation work; Creep; hmax = maximal indentation depth.  What are the difference and significant among HM, HV, and EIT?   The writing style like a lab report, while tests were carried out without objective, control groups, and the meaning of each data or the trend.  The general reads deserve more gaining after reading this manuscript.

4.        The artificial saliva should be specified either the brand name for commercial product or the detail compositions if prepared in the lab. In addition, why the 24 h aging was carried out in DDW, but the 3 months aging was in artificial saliva?

5.        In Page 4, Line 158, it was claimed that “Specimens were exposed to a 158 quasi-static indentation test according to ISO 14577”.  However, ISO 14577-1:2015 is for metallic materials, please confirm the suitability of referring this ISO standard?

6.        The conclusion of “Detailed laboratory characterization and comparison with 24 similar materials suggest similar clinical behavior” made without adequate evidence-based support data. 

1. In Title: composits è composites; behaviour è behavior

Author Response

All comments to the corresponding author have been addressed independently below. The authors’ rebuttal is always in BLUE and where changes have been added to the revised manuscript in light of the reviewer's comments these are presented in RED.

The author would firstly like to thank the reviewer for taking the time to read and critically appraise the manuscript and secondly to thank the reviewers for their positive constructive comments in improving the work.

Comments and Suggestions for Authors

Reviewer 1 comments:

The objective of this study is to investigate the dental composites with reversible addition-fragmentation-chain transfer mediated polymerization: An in-vitro comparison of their elasto-plastic and viscoelastic behaviour.  Some suggestions are listed as following:

  1. The title of this manuscript is ambiguous to general readers in many aspects.  First, this manuscript claimed the term of “reversible addition-fragmentation-chain transfer mediated polymerization” only by testing the mechanical properties of 2 commercial dental composite products but without showing anymechanism of chemical reaction related with “reversible addition−fragmentation” and “chain transfer polymerization” using monomers and β-allyl 95 sulfone.  In addition, this type of writing style should only happen in author conduct your own RAFT polymerization with some modified monomers or reagents.  Or the manuscript is some kinds of review paper.

Author’s response:  I can understand the reviewer's concern, but the article is about a special dental materials category and is intended for researchers in the field.  As explained in the introduction, the RAFT polymerization was introduced into dental materials very late and there are only a few, in fact only two such materials. So, RAFT-mediated polymerization characterizes them as a special category of composites and they are identified in dentistry by this description.  That's right, I didn't develop the materials and the article doesn't analyze the mechanism of RAFT polymerization. It describes a category in dental materials that should be recognizable.

I changed the title to start with what was done in the work (elastoplastic and viscoelastic behavior) and placed in the secondary level the designation of the material.

“An in-vitro comparison of elastoplastic and viscoelastic behaviour of dental composites with reversible addition-fragmentation-chain transfer mediated polymerization”.

  1. Second, the term of “in-vitro comparison” usually refers to a test occurring in a laboratory involving the studying of the biological effects for microorganisms or human/animal cells in culture.  I don’t see any relevant in vitro study results in this manuscript.

Author’s response:  I disagree that in-vitro refers only to laboratory studies in the field of microbiology. The term “in-vitro” means nothing else than outside the living body and in an artificial environment. And that is what has been done in this study. Please note that the term is often used as such in dental and medical research of any kind.

  1. Third, the term of “elasto-plastic and viscoelastic behavior” should be clearly defined and classified for dental composites by testing their stress-strain curves or any other appropriate test according their definition. Elastoplasticity is a phenomenon where a material undergoes permanent deformation when the applied stress exceeds a critical value, and elastoplastic parameters are quantities that describe the elastic and plastic deformation behavior of a material.  There are many terms and data showed in the Table 3 like HM = Martens hardness, HV = Vickers hardness; EIT = indentation modulus; μIT = We / Wt; We = elastic indentation work; Wt = total indentation work; Creep; hmax = maximal indentation depth.  What are the difference and significant among HM, HV, and EIT?   The writing style like a lab report, while tests were carried out without objective, control groups, and the meaning of each data or the trend.  The general reads deserve more gaining after reading this manuscript.

Author’s response:  The parameters mentioned are explained in detail in the "Material and Methods" section, the standard used for the measurements is also given and the parameters are very well-known in dental research.

Please consider the manuscript text “Within each load-unload-cycle, the load (F) and indentation depth (h) of the indenter were continuously measured, allowing calculating a range of parameters that characterises the elastic and plastic deformation. The integral of the force with depth (= Fdh) defines the total mechanical work of indentation Wtotal. During the indentation procedure a part of the total mechanical work is consumed as plastic deformation work Wplast, while the rest is set free as work of the elastic reverse deformation Welastic. The ratio of the elastic reverse de-formation work of indentation (Welast) to the total mechanical work of indentation (Wtotal) was then calculated and it represents a prerequisite variable for the further DMA test (We-last/Wtotal = µIT). Further parameters were then determined from the load-indentation depth variation; these include the indentation modulus EIT, which was calculated from the slope of the tangent of the indentation depth curve at the maximum force. Hardness with its plastic and elastic components was calculated by evaluating the impression created during the indentation. For this purpose, the projected indenter contact area (Ac) was determined from the force–indentation depth curve, while considering the indenter correction based on the Oliver and Pharr model and described in ISO 14577 [15] and a previous calibration with sapphire and quartz glass. The resistance to plastic deformation only is described by the indentation hardness (HIT = Fmax/Ac) and its more familiar correspondent, the Vickers hardness (HV = 0.0945 × HIT). The universal hardness (or Martens hardness = F/As(h)) was calculated by dividing the test load by the surface area of the indentation under the applied test load (As) and characterizes both plastic and elastic deformation. Creep was calculated from changes in indentation depth during the 5 s of maintaining maximal indentation force during the indentation process described above. The indentation depth at maximal force is also indicated (hmax)”.

  • The artificial saliva should be specified either the brand name for commercial product or the detail compositions if prepared in the lab. In addition, why the 24 h aging was carried out in DDW, but the 3 months aging was in artificial saliva?

Author’s response:  Thank you for this important note. The artificial saliva is made at the university.  I have detailed the composition.  “artificial saliva (pH 6.9; composition: 1.2 g potassium chloride, 0.84 g sodium chloride, 0.26 g di-potassium hydrogen phosphate and 0.14 g calcium chloride dihydrate per 1,000 g of water) “

About the type of storage: I used distilled water for the 24-hour storage, as this is prescribed in the dental ISO standard.  The mechanical parameters are measured after a 24-hour post-polymerization in distilled water at 37°C.  In addition, I tested aging behavior by simulating clinically relevant conditions using artificial saliva.

  1. In Page 4, Line 158, it was claimed that “Specimens were exposed to a 158 quasi-static indentation test according to ISO 14577”.  However, ISO 14577-1:2015 is for metallic materials, please confirm the suitability of referring this ISO standard?

Author’s response:  Hardness measurements were historically developed for metals and then applied to all other materials such as ceramics, polymers, etc. The ISO standard is based on the theory of Oliver and Pharr and is used in research for all types of materials including biological substrates. For more details, please see the work of the above two authors, among many other research papers.

  1. The conclusion of “Detailed laboratory characterization and comparison with 24 similar materials suggest similar clinical behavior” made without adequate evidence-based support data.

Author’s response:  This aspect is indeed debatable. The data are evidence-based insofar as parameters measured in the present study, such as strength or hardness, have indeed been related to the clinical success of a dental material. Please note the wording of this statement. I used the word "suggests".  For the avoidance of doubt, I have limited the last sentence of the abstract to the measured facts and changed the sentence to " Detailed laboratory characterization indicates comparable in-vitro behavior with clinically successful materials”.

Reviewer 2 Report

No „I” or “we” exists in the scientific language. Such expressions should have been written either in passive voice or impersonal form.

Please re-write the abstract in order to gain it a well-structured, more understandable form, as described in the Instructions for Authors. In its current state, it shines as chaotic and does not invite readers to deepen themselves into the full text.

Lines 76-97 should be transferred into the discussion. There are better places to discuss the advantages and disadvantages than the introduction. What is more, I kindly ask You to discuss in more detail the clinical efficiency of such materials.

Why are some table cells in Table 1 in bold?

Why were some tests performed only after 24 hours, while others only after three months? There is no clear explanation for this in the text. For this reason, the information contained in the text is incoherent. E.g., SEM analysis could have been performed after three months.

I found the rest of the discussion and the conclusions useful. However, I would like to ask you to change the conclusions to be listed in points.

I found the quality of English language fairly good.

Author Response

All comments to the corresponding author have been addressed independently below. The authors’ rebuttal is always in BLUE and where changes have been added to the revised manuscript in light of the reviewer comments these are presented in RED.

The author would firstly like to thank the reviewers for taking the time to read and critically appraise the manuscript and secondly to thank the reviewers for their positive constructive comments in improving the work.

Comments and Suggestions for Authors

Reviewer 2 comments:

No „I” or “we” exists in the scientific language. Such expressions should have been written either in pa ssive voice or impersonal form.

 Author’s response:  I thank the reviewer for this important point, with which I agree. In fact, the word “we” was used once in the abstract based on the observation that this is often how the abstract is presented in MDPI journals. I have changed this accordingly.

Please re-write the abstract in order to gain it a well-structured, more understandable form, as described in the Instructions for Authors. In its current state, it shines as chaotic and does not invite readers to deepen themselves into the full text.

Author’s response: Thank you for your feedback on the lack of clarity in the abstract. It actually reflected the line of the analyzed parameters and results. It has been modified for a more appealing look; Please note the changes in the manuscript text.

Lines 76-97 should be transferred into the discussion. There are better places to discuss the advantages and disadvantages than the introduction. What is more, I kindly ask You to discuss in more detail the clinical efficiency of such materials.

Author’s response:  Following your recommendation, the above paragraph has been moved to the discussion. Regarding the clinical efficiency of the RAFT-mediated RBCs, there is no data available so far. The reason why the RAFT-mediated polymerization was introduced is explained in the discussion as the development pursues the idea of optimizing polymerization in the depth of a sufficiently opaque material, or aimed at faster and more efficient polymerization. Please consider the 3-rd paragraph of the discussion.

Why are some table cells in Table 1 in bold?

Author’s response:  Only the heading of the table was in bold. This has now been changed in all tables so that no bold cells are visible.

Why were some tests performed only after 24 hours, while others only after three months? There is no clear explanation for this in the text. For this reason, the information contained in the text is incoherent. E.g., SEM analysis could have been performed after three months.

Author’s response:  A 24-hour storage was employed as it is prescribed in the dental materials ISO standard, which was cited in materials and methods. Aging is an addition to simulate clinical conditions. Also in accordance with a unanimously accepted procedure, the SEM was performed on samples aged for 24h, as we intend with this examination to observe the shape, distribution, and structure of the filler.

I found the rest of the discussion and the conclusions useful. However, I would like to ask you to change the conclusions to be listed in points.

Author’s response:  Thank you for your comments and appreciation. I have changed the conclusions as recommended.

Reviewer 3 Report

Dear Authors,

Your article is fascinating and well-written. I suggest reducing the length of the introduction and discussion paragraphs to give greater prominence to the results. 

The English used is good. The text is clear and easily understandable. Only minor changes are required before publication.

Author Response

All comments to the corresponding author have been addressed independently below. The authors’ rebuttal is always in BLUE and where changes have been added to the revised manuscript in light of the reviewer's comments these are presented in RED.

The author would firstly like to thank the reviewers for taking the time to read and critically appraise the manuscript and secondly to thank the reviewers for their positive constructive comments in improving the work.

Comments and Suggestions for Authors

Reviewer 3 comments:

Dear Authors,

Your article is fascinating and well-written. I suggest reducing the length of the introduction and discussion paragraphs to give greater prominence to the results. 

Author’s response: Thank you for your appreciation! The introduction has been shortened by moving Lines 76-97 into the discussion.

Comments on the Quality of English Language

The English used is good. The text is clear and easily understandable. Only minor changes are required before publication.

Author’s response:  Thank you for your comments.  I carefully reviewed the paper and made appropriate corrections.

Round 2

Reviewer 1 Report

1. I disagree author's disagreement in  “in-vitro” means nothing else than outside the living body and in an artificial environment".   Generally speaking, the term of in vivo, ex vivo, and in vitro  are used with clear definition related with cells.    The mechanical tests in this manuscript was not relvent.

2. If the term of RFAT only for describing a category in dental materials, then 80% of your content in introduction is related with RFAT might no be suitable.

3. To compare the aging effect for 24 hr and 3 months in different surrounding (DDW vs. artificial saliva) is questionable in term of experimental design.

4. Some guessing from authors could be listed in discussion section but not suitable in conclusion section.

The English is OK!

Reviewer 2 Report

Dear author

Thank You for Your response and the changes made.

Although you made some corrections, I found it difficult to find them as You did not mark the changes within the text in red.

1. First, the website of the journal clearly states written:

"Abstract: The abstract should be a total of about 200 words maximum. The abstract should be a single paragraph and should follow the style of structured abstracts, but without headings: 1) Background: Place the question addressed in a broad context and highlight the purpose of the study; 2) Methods: Describe briefly the main methods or treatments applied. Include any relevant preregistration numbers and species and strains of any animals used; 3) Results: Summarize the article's main findings; and 4) Conclusion: Indicate the main conclusions or interpretations. The abstract should be an objective representation of the article: it must not contain results which are not presented and substantiated in the main text and should not exaggerate the main conclusions."

Please follow the instructions in the journal.

2. Why were some tests performed only after 24 hours, while others only after three months? There is no clear explanation for this in the text. For this reason, the information contained in the text is incoherent. E.g., SEM analysis could have been performed after three months.

Author's response: A 24-hour storage was employed as it is prescribed in the dental materials ISO standard, which was cited in materials and methods. Aging is an addition to simulate clinical conditions. Also, in accordance with a unanimously accepted procedure, the SEM was performed on samples aged 24h, as we intend with this examination to observe the shape, distribution, and structure of the filler.

I found Your explanation useful. However, it still needs to be clarified in the text of the manuscript. It looks as if You made some test at this time point and the other at another only because You wished to do so. The reader needs a brief explanation of why and when something was done. Otherwise, it can leave the impression that the study was based; Please apply some description of the way of conducting the experiment in the materials and methods section, similarly, but more in detail that You have written in the response.

Reviewer 3 Report

Dear Authors,

thank you for your work. The required changes r were made and the manuscript was improved. 

The paper is well written and the English language is appropiate.